

# Night of the hunter: using cameras to quantify nocturnal activity in desert spiders

Tamara I. Potter[1,2], Aaron C. Greenville[2,3] and Christopher R. Dickman[2,3]

[1] Terrestrial Ecosystem Research Network, School of Biological Sciences, University of Adelaide, Adelaide, South Australia, Australia

[2] Desert Ecology Research Group, School of Life and Environmental Sciences, University of Sydney, Sydney, New South Wales, Australia

[3] National Environmental Science Program Threatened Species Recovery Hub, School of Life and Environmental Sciences, University of Sydney, Sydney, New South Wales, Australia

## ABSTRACT

Invertebrates dominate the animal world in terms of abundance, diversity and biomass, and play critical roles in maintaining ecosystem function. Despite their obvious importance, disproportionate research attention remains focused on vertebrates, with knowledge and understanding of invertebrate ecology still lacking. Due to their inherent advantages, usage of camera traps in ecology has risen dramatically over the last three decades, especially for research on mammals. However, few studies have used cameras to reliably detect fauna such as invertebrates or used cameras to examine specific aspects of invertebrate ecology. Previous research investigating the interaction between wolf spiders (Lycosidae: *Lycosa* spp.) and the lesser hairy-footed dunnart (*Sminthopsis youngsoni*) found that camera traps provide a viable method for examining temporal activity patterns and interactions between these species. Here, we re-examine lycosid activity to determine whether these patterns vary with different environmental conditions, specifically between burned and unburned habitats and the crests and bases of sand dunes, and whether cameras are able to detect other invertebrate fauna. Twenty-four cameras were deployed over a 3-month period in an arid region in central Australia, capturing 2,356 confirmed images of seven invertebrate taxa, including 155 time-lapse images of lycosids. Overall, there was no clear difference in temporal activity with respect to dune position or fire history, but twice as many lycosids were detected in unburned compared to burned areas. Despite some limitations, camera traps appear to have considerable utility as a tool for determining the diel activity patterns and habitat use of larger arthropods such as wolf spiders, and we recommend greater uptake in their usage in future.

Corresponding author
Tamara I. Potter,
tamara.potter@adelaide.edu.au

# INTRODUCTION

Invertebrates dominate the world's animal biota in terms of abundance, diversity and biomass (*Black, Shepard & Allen, 2001*; *Ellwood & Foster, 2004*; *McCollough, 1997*). They play a critical role in maintaining ecosystem function by performing basic services

such as energy and nutrient cycling, pollination, herbivory and seed dispersal (*Black, Shepard & Allen, 2001*; *Freckman et al., 1997*; *Oberprieler, Andersen & Braby, 2019*). Some invertebrates are also keystone species and are fundamental in regulating the structure of biotic communities (*Black, Shepard & Allen, 2001*). However, despite their obvious ecological importance, there is a disproportionate focus in many areas of research on vertebrates, with a clear disparity in regards to our knowledge and understanding of the ecology of invertebrate fauna (*Oberprieler, Andersen & Braby, 2019*; *Ponder & Lunney, 1999*).

Many methods are employed to sample terrestrial invertebrates. These can be either direct or indirect, and vary depending on the aims of the study, the habitat and often the habits of the taxa being sampled. For instance, one of the most common methods to efficiently sample ground-dwelling invertebrates is the wet pitfall trap (*Callan et al., 2011*; *Gist & Crossley, 1973*; *Potter, Greenville & Dickman, 2018*). Invertebrates can also be collected as by-catch from vertebrate pitfall traps (*Oberprieler, Andersen & Braby, 2019*; *Potter, Greenville & Dickman, 2018*; *Woinarski et al., 2002*), while netting, vacuuming and beating of plant leaves and fronds are used to sample invertebrates that live within vegetation (*Callan et al., 2011*; *Popic, Davila & Wardle, 2013*; *Southwood & Henderson, 2000*). Other methods include hand netting and pan traps for aerial invertebrates (*Popic, Davila & Wardle, 2013*), raking leaf litter (*Callan et al., 2011*), fogging (for arthropods that dwell in the forest canopy; *Ellwood & Foster, 2004*), and diurnal or nocturnal hand searching and collecting (*Callan et al., 2011*; *Gobbi et al., 2018*; *Potter, Greenville & Dickman, 2018*). One of the key benefits of these approaches is that they collect actual animals, permitting detailed examination of morphology and providing material for genetic analyses, thus facilitating taxonomic identification (*Wong, Guénard & Lewis, 2019*). These methods also allow information to be gathered on abundance, diversity and population statistics (e.g., sex ratio or age class). One prominent downside, however, is that these direct methods usually result in the death of individuals captured and thus removes invertebrates from the study system. In comparison, surveys of burrows or nests, tracks or other traces, and records of vocalisations, are examples of non-invasive, indirect gauges of arthropod presence or activity that do not disrupt local population abundances (*Henschel, 2002*; *Nørgaard, Henschel & Wehner, 2006*; *Southwood & Henderson, 2000*).

The use of remote-sensing camera traps has increased dramatically in the last three decades due to the numerous advantages they provide in comparison with traditional sampling methods (*Meek et al., 2015*; *Potter, Brady & Murphy, 2019*). While cameras are relatively non-invasive, collect information on numerous species, are convenient and cost-effective, their most valuable attribute is that they can monitor continuously over extended periods of time and can be utilised at a range of spatial scales (*De Bondi et al., 2010*; *Harley et al., 2014*). Cameras have been employed to observe cryptic fauna (*Claridge et al., 2004*; *Nelson, Scroggie & Belcher, 2014*; *Potter, 2017*), provide data to estimate occupancy and abundance (*Gowan & Vernes, 2014*; *Rowcliffe et al., 2014*), monitor animal behaviour (*Vernes, Smith & Jarman, 2014*), and examine diel activity patterns (*Diete et al., 2017*; *Meek, Zewe & Falzon, 2012*). However, due to limitations regarding trigger mechanisms (e.g., heat or movement signatures), cameras have largely been used in research on large and small

mammals, with fewer studies using cameras to detect birds, reptiles, amphibians and, especially, invertebrates (*Agha et al., 2018*; *Collett & Fisher, 2017*; *Hobbs & Brehme, 2017*; *Lortie, Amber & Anya, 2012*). Although the number of studies using cameras to monitor reptiles is on the rise (*Bluett & Cosentino, 2013*; e.g., *Pagnucco, Paszkowski & Scrimgeour, 2011*; *Welbourne et al., 2020*), there are still very few researchers using this tool to explore any aspects of invertebrate ecology (*Agha et al., 2018*).

Previous research investigating the interaction between wolf spiders (Lycosidae: *Lycosa* spp.) and a small marsupial, the lesser hairy-footed dunnart (*Sminthopsis youngsoni*), resulted in the knowledge that (1) camera traps provide a viable method for detecting wolf spiders and (2) that captured images can be used to investigate temporal activity patterns and species interactions (*Potter, Greenville & Dickman, 2017a*; *Potter, Greenville & Dickman, 2017b*; *Potter, Greenville & Dickman, 2017c*; *Potter, Greenville & Dickman, 2017d*; *Potter, Greenville & Dickman, 2018*). Here, we re-examine lycosid activity to determine whether these patterns vary with different environmental conditions. Additionally, we extended our study to assess the capability of cameras to detect other invertebrate taxa. Our focus was primarily on lycosids as these active hunting spiders indicate the presence of diverse smaller invertebrates that form their major prey (*Nyffeler & Benz, 1987*) and because they are often selected as preferred prey by small vertebrate predators (*Potter, Greenville & Dickman, 2018*), and thus are important components of their constituent communities. Because of the ubiquity and importance of lycosids in trophic webs in many arid environments (*Henschel, 1998*; *Punzo, 2003*; *Russell-Smith, Ritchie & Collins, 1987*), we focused our sampling in the Simpson Desert of central Australia. To test the utility of cameras in distinguishing lycosid activity under different environmental conditions, we deployed camera traps on dune crests and dune bases (swale) in recently burned and unburned habitats. Owing to a high abundance of small, crepuscular marsupials in the study region that include spiders and other arthropods in their diet (*Fisher & Dickman, 1993a*; *Fisher & Dickman, 1993b*), we expected that lycosids would be less active near dawn and dusk in open habitats (dune crests and burned habitat) than in sheltered habitats (dune bases and unburned habitat) to reduce their risk of predation.

## MATERIALS & METHODS

### Study site

This study was undertaken over an area of 0.06 km$^2$ near Main Camp, on Ethabuka Reserve (23°46′S, 138°28′E), in the north-eastern Simpson Desert, Queensland, Australia, between July and October 2016. Access to the reserve was provided by Alex Kutt, Bush Heritage Australia Regional Ecologist, and Matt and Amanda Warr, the Reserve Managers, at the time of the study. The Simpson Desert is characterised by long, parallel sand dunes that run north-northwest to south-southeast, are 0.6–1 km apart and can be up to 10 m high (*Dickman et al., 2010*; *Kwok et al., 2016*; *Purdie, 1984*). The dominant vegetation is spinifex (*Triodia basedowii*) grassland; however, the dune crests are sparsely covered in shrubs and other perennial species (*Greenville et al., 2009*; *Kwok et al., 2016*). Small stands of gidgee trees (*Acacia georginae*), mallee eucalypts (*Eucalyptus* spp.) and other *Acacia* shrubs occur

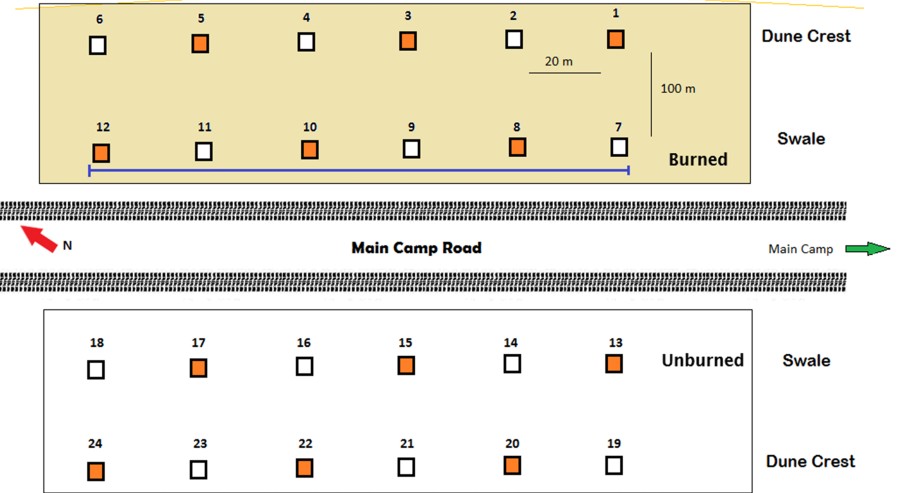

**Figure 1 Arrangement of remote-sensing cameras near Main Camp, Simpson Desert, Queensland.** Twenty-four cameras were deployed, half in recently burned (summer of 2011/2012) and half in unburned habitat. Within these two habitats, six cameras were situated on the dune crest and six in the swale. Two camera positions were tested: (A) angled at 45° (orange squares, $n = 12$) and (B) vertically (white squares, $n = 12$). The purple line indicates the transect followed during spider spotlighting surveys.

on the heavier clay soils of the interdune swales (*Greenville et al., 2016*; *Kwok et al., 2016*; *Wardle et al., 2015*). Wildfires occur commonly in the study region, ignited by lightning strikes during summer thunderstorms (*Greenville et al., 2009*; *Letnic & Dickman, 2006*). The most recent occurred over the summer of 2011/2012, with the majority of study grids on Ethabuka Reserve being patchily burned (*Greenville, 2015*; *Verhoeven et al., 2020*).

## Camera deployment

To investigate activity patterns of lycosids across the dune system and in relation to different environmental conditions, 24 Reconyx PC800 Hyperfire™ cameras (Reconyx, Inc., Holmen, WI, USA) were established in burned and unburned areas, as well as on dune crests and dune bases in the adjacent interdune swale (Fig. 1). Cameras were deployed for 98 days, with each camera positioned 0.5 m above the sand surface on metal posts set 20 m apart along four 100 m long transects running south-east to north-west (consistent with the prevailing dune direction). Additionally, 12 cameras were orientated at 45° to incorporate a greater field of view while the other 12 cameras were orientated vertically to increase the likelihood of capturing and identifying lycosids and other invertebrates (*Collett & Fisher, 2017*). Cameras were deployed from July to October 2016.

Camera settings included both time-lapse and motion-triggered images, with motion-trigger capturing single images in rapid-succession (i.e., no delay between trigger events) and sensitivity set to high to maximize detections. In time-lapse a single image was taken every 5 min from 19:00 h to 7:00 h (i.e., from dusk to dawn). Time-lapse was employed to increase captures of invertebrates as well as allowing for a controlled level of sampling effort across cameras, sites and time (*Hobbs & Brehme, 2017*). Time-lapse was not set to

operate by day, in part to conserve battery life, but largely because previous research has shown desert lycosids to be nocturnal, constraining their activity to relatively cool nights, and relying on burrows to retreat from the heat of the day (*Estrada, 2008*; *Framenau, 2015*). Pilot observations in this study and extended periods working in the study system during the day (e.g., a minimum of 85 h in targeted ground searches in 2016 and 2017) also confirmed zero lycosid activity by day (C Dickman, 2016–2017, unpublished data). Our nocturnal camera settings precluded captures of day-active invertebrates, such as some orthopterans, but diurnal invertebrates were not the main focus of our investigations.

Each photo was tagged with the following details: habitat (burned or unburned), location (crest or swale), position (angled or vertical), camera ID number and, if present, the fauna species and identification confidence level (possible, probable or definite). Only images with high confidence (definite and probable) in identification were used in analyses. Lycosids (two species of *Lycosa*; *Potter, Greenville & Dickman, 2018*) were distinguished from other ground-dwelling spiders such as prowling spiders (Family Miturgidae) by the raised carapace in images and rounded shape of the prosoma (Fig. 2). As there are no site-specific identification keys, lycosids were identified to the genus *Lycosa* using a field guide for spiders of Australia (*Whyte & Anderson, 2017*). Additionally, voucher specimens collected for this study are currently awaiting formal taxonomic resolution (R Raven, Queensland Museum, 2020, pers. comm.). All tags were written to the EXIF data of each file using the multi-format graphics program XnView MP v 0.83 (*Gougelet, 2016*).

## Spotlight surveys

To assess the efficacy of remote-sensing cameras in detecting and revealing activity patterns of lycosids, spotlighting surveys were also undertaken in October 2016 to provide data against which camera activity data could be compared. Spotlighting involved surveying for a total of 10 min every hour between 19:30 h (dusk) and 05:30 h (dawn) along a 100 m transect. Lycosid eyeshine was detected using a Fenix TK35 (960 lumens) hand-held spotlight (*Robinson & Thomson, 2016*). Lycosids were identified by their distinctive raised carapace and unique eye size and arrangement, which consists of four large posterior eyes that form a square on the carapace and four small anterior eyes that form a single row (*Framenau, Baehr & Zborowski, 2014*). In comparison, other common ground-dwelling spiders such as prowling spiders (Miturgidae) and ground-spiders (Gnaphosidae) have eight small eyes arranged in two rows of four (*Framenau, Baehr & Zborowski, 2014*). The number of lycosids observed in each 10-minute survey were tallied. Spotlighting was repeated over three nights (33 surveys in total), following the same 100 m transect each time for consistency and to reduce bias towards open areas where walking was more straightforward and spiders more easily detected (Fig. 1).

## Statistical analyses

The command line package 'exiftool' was used to extract EXIF data from each image and write it to an Excel file. Image time stamps were examined to determine independence, with photographs likely to be of the same individual removed (i.e., those less than 30 mins apart).

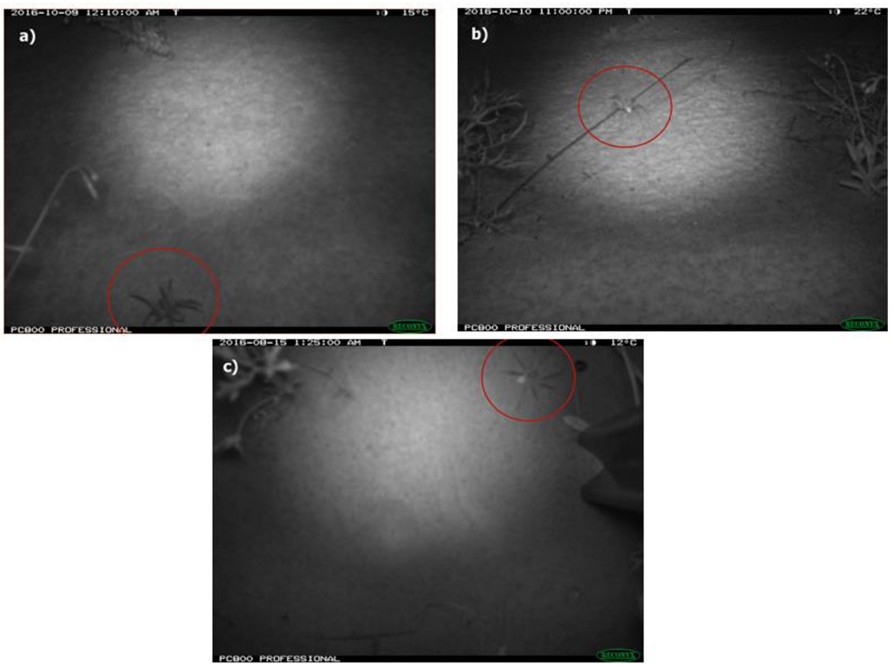

**Figure 2** **Photos of wolf spiders (Lycosidae: *Lycosa* spp.) taken using remote-sensing cameras deployed at Main Camp, Simpson Desert, Queensland between July and October 2016.** Two camera positions were employed to maximise capture success of the study species, i.e., 45° angle (A & B) or vertical (C). Lycosids were distinguished from other ground-dwelling spiders (e.g., prowling spiders: Miturgidae) by the raised carapace (evident in image A) and shape of their prosoma, which is more rounded compared to miturgids.

In initial analyses, to determine the general activity patterns of lycosids across the dune system, a species detection map was generated using the package 'camtrapR' v 2.0.3 in R v 4.0.2. Following this, general lycosid diel activity was ascertained from images pooled across all cameras, and the circular statistics program Oriana v 4.02 (*Kovach Computing Services, 2013*) was used to calculate mean activity times and 95% confidence intervals. These statistics were also calculated for spotlighting data. No statistical tests could be made between the two distributions (camera and spotlighting), as spotlighting data were grouped into time classes. Consequently, means and confidence intervals (CI) were compared and significance determined if the 95% confidence intervals did not overlap. In subsequent analyses, to determine if lycosid activity patterns varied under different circumstances, nocturnal activity patterns from cameras in burned and unburned habitats were compared, as were those from dune crests and bases, using the Mardia-Watson-Wheeler Test (*Fisher, 1993*; *Mardia, 1969*; *Mardia & Jupp, 2000*) in Oriana. Also known as the Uniform Scores Test, this is a non-parametric test for determining whether two or more circular distributions are identical (*Fisher, 1993*; *Mardia & Jupp, 2000*; *Tasdan & Yeniay, 2014*).

**Table 1** **Invertebrate taxa captured on remote-sensing cameras in different habitats in the Simpson Desert.** Data are raw numbers of images for all taxa except lycosids, which are independent records. Raw images are used as independence between records is difficult to establish for some taxa (e.g., ants and moths). Only images with high confidence in identification ('definite') are presented here.

| Taxa | No. of images | Dune Crest | Dune Base | Burned | Unburned |
|---|---|---|---|---|---|
| Ant (Hymenoptera) | 1,215 | 624 | 591 | 501 | 714 |
| Beetle (Coleoptera) | 769 | 565 | 204 | 214 | 555 |
| Grasshopper (Orthoptera) | 5 | 3 | 2 | 1 | 4 |
| Lycosid (Lycosidae) | 155 | 20 | 95 | 30 | 85 |
| Moth (Lepidoptera) | 69 | 55 | 14 | 40 | 29 |
| Scorpion (Scorpiones) | 1 | 1 | 0 | 1 | 0 |
| Other spiders (Arachnidae) | 27 | 22 | 5 | 1 | 26 |
| **Total** | **2,356** | **1,358** | **998** | **841** | **1,515** |

# RESULTS

## Overall activity

Overall, 479,210 images were taken during the study period. Processing took around 100 h with about 1.6% of images comprising fauna, including 2,356 images of 7 invertebrate taxa made with high identification confidence (Table 1; Fig. 3). There were an additional 2,747 raw images with an additional 7 taxa identified, but confidence in identification was lower. Invertebrates were mainly captured in time-lapse imagery or otherwise in the background of images triggered by vertebrate species; i.e., invertebrates did not trigger motion capture. There were 155 records of lycosids from camera images and 352 lycosids were recorded during spotlighting surveys. Spotlighting data showed lycosids to be active throughout the night, with mean activity of 00:19 h (95% CI [23:58–00:40 h]; Fig. 4). In comparison, camera data revealed a mean activity time of 23:09 h (95% CI [22:35–23:42 h]; Fig. 4).

## Environmental influences

Greater numbers of invertebrates were recorded on dune crests compared to dune bases, and in unburned compared to burned habitats (Table 1). Additionally, more invertebrates were captured on angled cameras (1516 images) than on vertically oriented cameras (840 images). With respect to lycosids, more were recorded on cameras angled at 45° ($n = 123$), while only 32 images of lycosids were captured on cameras positioned vertically. Activity was slightly higher on dune bases, with 87 images compared to 68 records on dune crests. However, the detection map revealed that activity on dune bases was dominated by a single camera (camera 16) which logged 65.5% of lycosid records in this habitat (Fig. 5). More lycosids were detected in unburned habitats than in burned habitats (102 and 53 records respectively). No difference was detected in diel activity patterns between dune crests and dune bases ($W = 2.61$, $p = 0.27$, Fig. 6), with the mean activity on dune crests occurring at 23:34 h (95% CI [21:45–23:22 h]) and mean activity on the dune base at 23:37 h (95% CI [22:52–00:23 h]). Similarly, no significant difference was detected in the diel patterns of lycosids in burned and unburned habitats ($W = 2.34$, $p = 0.31$, Fig. 7), with mean activity

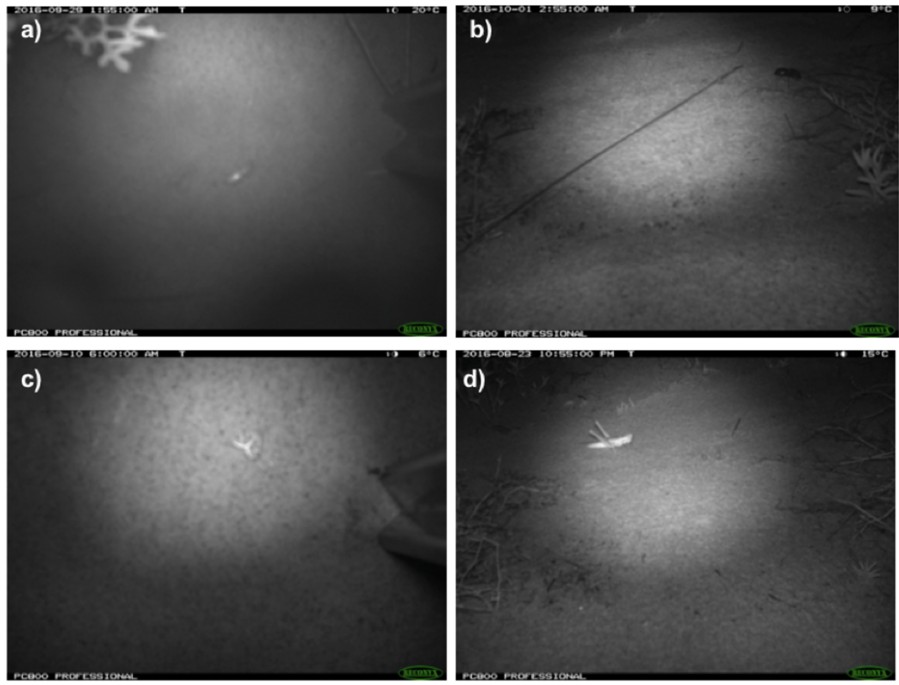

**Figure 3** **Photos of invertebrate taxa from remote-sensing cameras.** Invertebrate taxa captured in time-lapse images from cameras deployed at Main Camp, Simpson Desert, Queensland between July and October 2016: (A) and (B) beetles (Coleoptera), (C) moth (Lepidoptera), and (D) grasshopper (Orthoptera).

at 23:53 h (95% CI [22:47–00:58 h]) for burned habitat and 22:49 h (95% CI [22:11–23:27 h]) for unburned habitats.

## DISCUSSION

The results we gathered confirm the value of camera traps in revealing activity patterns of desert-dwelling lycosids. Despite capturing lycosids only using the time-lapse setting, sufficient images were obtained to permit detailed analyses of lycosid temporal and spatial activity across the landscape. Additionally, other invertebrate taxa were confidently identified in over 2,300 images. We first examine the context of these results before discussing the limitations of the study and future directions. Overall, camera traps appear to have considerable utility as a tool for ecological investigations and longer-term monitoring of invertebrates.

### Diel activity patterns

Lycosids were found in camera images to be most active 9 min after 23:00 h. Despite the mean activity time from spotlighting occurring at a similar time as trends from camera data (about an hour later, at 19 min past midnight), there was a clear distinction in the spread of activity times between these two survey techniques. Although there are minor fluctuations in activity, in general spotlighting revealed lycosids to be active right through the night. In contrast, camera data showed a distinct peak in activity just after dusk (19:00–21:00),
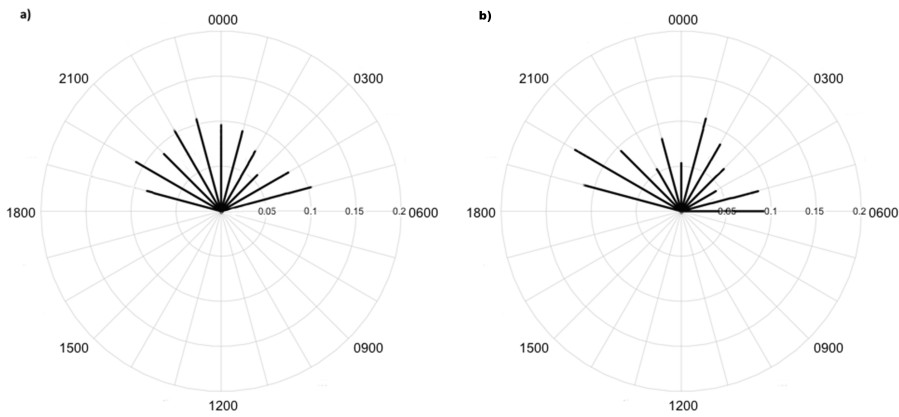

**Figure 4** **Diel activity patterns of lycosids from spotlighting and camera data.** Diel activity patterns of lycosids based on (A) spotlighting data and (B) data pooled across all cameras at Main Camp, Simpson Desert, Queensland. Activity is the proportion of records aggregated in each hourly period.

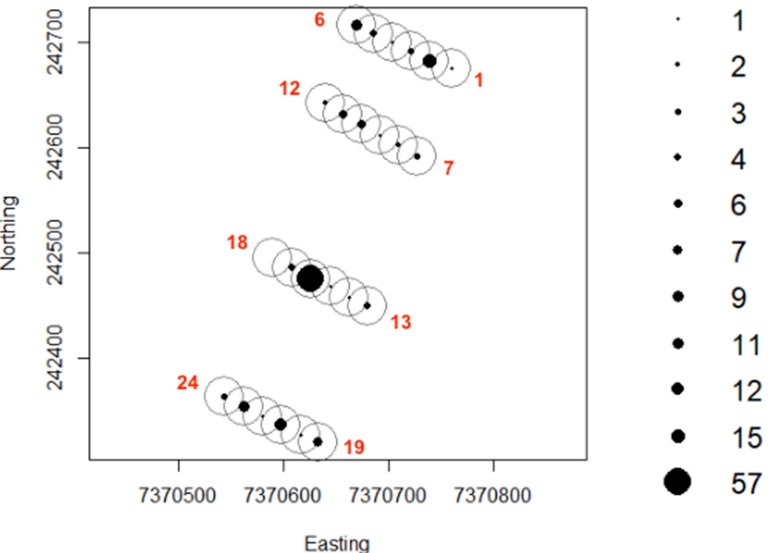

**Figure 5** **Lycosid detection map generated using camera records.** Camera numbers are indicated in red. Cameras 1 through to 6 and 19 to 24 are positioned on dune crests while 7 to 12 and 13 to 18 are in the interdune swale. Additionally, the first 12 cameras are in burned habitat with the remaining 12 cameras in unburned habitat.

followed by lower activity during the remainder of the night. A peak in activity just after dusk was also observed in unburned habitat and dune bases, while greater spread in activity throughout the night was observed in burned habitats and on dune crests.
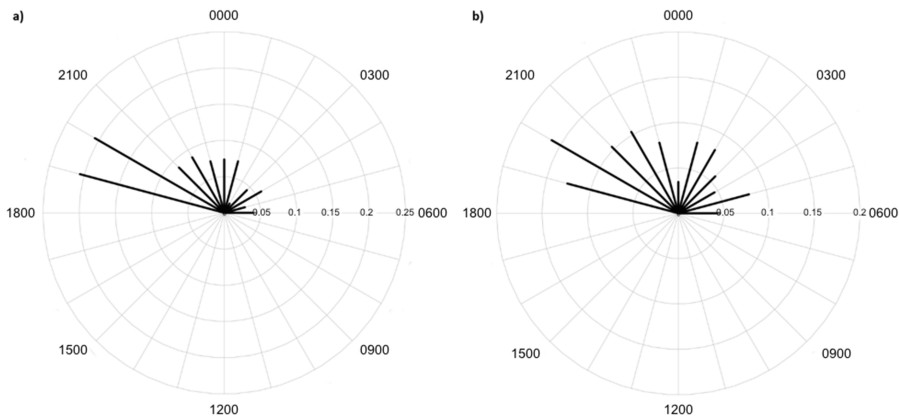

**Figure 6** **Diel activity patterns of lycosids on (A) dune crests and (B) dune bases using data extracted from camera images deployed at Main Camp, Simpson Desert, Queensland between July and October 2016.** Activity is the proportion of records aggregated in each hourly period.

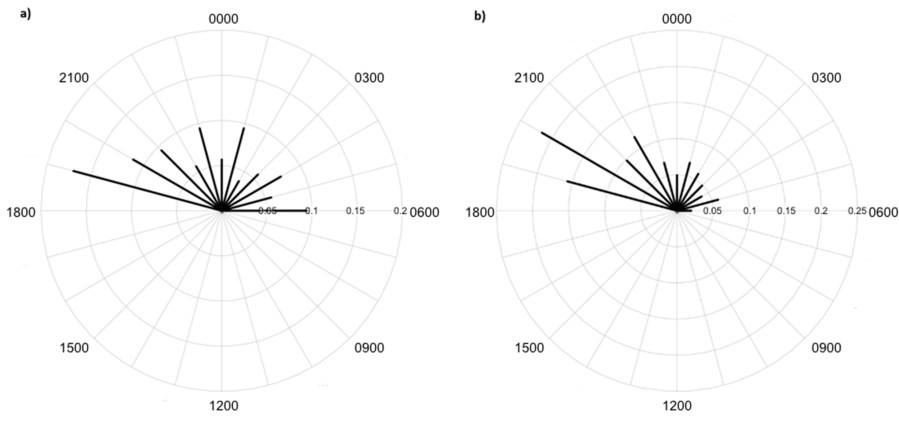

**Figure 7** **Diel activity patterns of lycosids in (A) burned and (B) unburned habitats using data extracted from camera images deployed at Main Camp, Simpson Desert, Queensland between July and October 2016.** Activity is the proportion of records aggregated in each hourly period.

This pattern is also contrary to our hypothesis that predicted lycosids to be less active near dawn and dusk in open habitats, such as dune crests and burned areas, as this is when the risk of predation is higher (*Geiser, 1994*; *Körtner & Geiser, 2011*; *Potter, Greenville & Dickman, 2018*). However, twice the number of lycosids were recorded in unburned habitats with considerable ground cover compared to burned habitats, which suggests that predation may still be driving activity patterns but on a spatial scale rather than a temporal one. A key result of fire is reduced vegetation cover, particularly a reduction in spinifex which is the dominant vegetation in the study region, and which is highly flammable (*Greenville et al., 2009*). Lycosids have been found previously to prefer microhabitats with less bare ground and more spinifex cover, as it provides a valuable refuge from predators, such as the lesser hairy-footed dunnart (*Potter, Greenville & Dickman, 2018*).

This preference for areas of greater vegetation cover is also consistent with the notion of intraguild predation, whereby intense competition can lead to subordinate species varying their activity to reduce encounter rates with dominant competitors (*Schoener, 1974*; *Palomares et al., 1999*). In general, victims tend to exhibit spatial avoidance before displaying temporal partitioning (*Schoener, 1974*; *Polis, Myers & Holt, 1989*). Conversely, lycosids may be more active in unburned areas as there is greater availability of prey, as more beetles, grasshoppers and other spiders were recorded in unburned areas. Alternatively, it may be a combination of these two dynamics: predators and prey, that drive these spatial trends.

## Advantages of camera traps

In addition to providing information on lycosid activity, camera traps offered a cost-effective, non-invasive tool to survey other invertebrate taxa. For instance, seven taxa were confidently recorded, with an additional seven taxa detected but identified with lower confidence, thus demonstrating the potential utility of cameras as a tool for gauging invertebrate diversity, occupancy or species richness. Camera set-up could be targeted towards greater detection of invertebrates or form an additional component of vertebrate camera surveys, thus providing a more comprehensive and efficient survey technique.

Detections, or capture rates, could be increased by using camera stations baited with food or scent lures. However, the use of an attractant largely depends on the research project or questions in focus. For instance, although captures of more cryptic or rare species can be increased, baiting can also modify behaviour and species interactions; i.e., some individuals can become 'trap-happy' while others may subsequently avoid the area (*Gerber, Karpanty & Kelly, 2012*; *Mills et al., 2019*). Additionally, the activity patterns revealed may be artificial, as a lure can often bring animals into an area they may not otherwise frequent, or at different time periods (*Du Preez, Loveridge & Macdonald, 2014*; *Gerber, Karpanty & Kelly, 2012*; *Mills et al., 2019*). Capture rates can be improved passively by extending the survey duration or by placing cameras in areas where species are more likely to be active (e.g., near burrows, along or around logs or debris), or where there is evidence of activity such as in the vicinity of tracks, scats, animals pads or burrows.

Cameras may also be able to detect certain life history traits. In particular, and although not observed during this study, the well-known phenomenon of lycosids transporting their offspring on their body would be discernable from camera images. If there were enough records of this event, differences in reproductive cycles in various habitats or between desert boom and bust periods could be explored. Invertebrate size could also be examined if a ruler is positioned within the camera field of view (*Collett & Fisher, 2017*). Cameras with video capability could also be employed to investigate foraging behaviour, burrow use or speed.

## Limitations

Spotlighting is the technique of searching for nocturnal animals by using a beam of light to detect an animal's reflected eye shine (*Robinson & Thomson, 2016*), and has become an accepted standard for surveying populations of arboreal and ground-dwelling

fauna, notably vertebrates (e.g., *Catling, Burt & Kooyman, 1997*; *Engeman & Vice, 2001*; *Wilmott et al., 2019*). Despite being a well-established technique, spotlighting is not always successful and factors such as dense vegetation, poor weather conditions (rain, fog), and the nature of the fauna species being targeted (e.g., cryptic, poor eye shine), can decrease the capacity for detections (*Catling, Burt & Kooyman, 1997*). Additionally, compared to cameras, spotlighting can be more disruptive to animal behaviour (*Robinson & Thomson, 2016*), is more labour-intensive, and yields results for single time-periods only. However, the successful use of spotlighting in this study was largely due to the open habitats of the Simpson Desert, the exceptionally bright eye shine of lycosids (*Robinson & Thomson, 2016*), the combination of 'search and pursuit' and 'sit and move' foraging tactics exhibited by individual lycosids (T Potter, 2016, pers. obs.),  as well as the high number of transect surveys completed (33 in total).

Although cameras are non-invasive, cost-effective and provide continuous data over extended periods of time, the greatest drawback in terms of this study related to the trigger mechanism. Most cameras use a passive infrared (PIR) sensor or heat-in-motion, which detects the presence of an animal when movement or a temperature differential occurs (*Meek et al., 2015*; *Rovero et al., 2013*; *Swann & Perkins, 2014*). Generally, Reconyx™ cameras (cameras used in this study) will be unreliable if the body temperature of an individual is within 5 °C of the ambient temperature (*Welbourne, 2014*). Therefore, animals that have a smaller heat signature, such as ectotherms whose body temperatures seldom fluctuate more than 3 °C from their surroundings, or smaller bodied organisms, are less likely to trigger the camera (*Glen et al., 2013*; *Harlow et al., 2010*; *Tobler et al., 2008*). Consequently, lycosids were captured only in time-lapse photos. The need for time-lapse imagery meant that a huge number of images was obtained which then had to be sorted and individually tagged. This was an arduous, time-consuming process, with 98.4% of images containing no fauna. Due to the high number of images, the potential for human error to overlook or mis-identify images was also relatively high. One way to address this problem may be to employ automated identification software, that even at its most basic, removes false-triggers or images that are devoid of fauna (*Hobbs & Brehme, 2017*; *Swinnen et al., 2014*). Nonetheless, this process is still fraught with challenges including lack of training data, accounting for environmental factors, such as wind and clouds that generate shifting light conditions and vegetation movement, and an accuracy level that is high enough to detect smaller invertebrates, as opposed to larger, more obvious mammals, birds or reptiles (*Hobbs & Brehme, 2017*; *Swinnen et al., 2014*). Consequently, the use of this particular technology is still in the early development stages. However, this study shows that remote cameras can be successfully used to survey some arthropod groups, and thus we hope it drives the collection of more images that could be used as training data for automatic methods.

Another option to improve detectability and efficiency is to reconsider the camera technology, namely the trigger mechanism (*Meek & Pittet, 2014*; *Rovero et al., 2013*). Hobbs' Active Light Trigger (HALT) is a tool that has been demonstrated to improve the detectability of small animals (*Hobbs & Brehme, 2017*). This technology uses a 3 mm near-infrared (NIR) optical beam that is positioned above an angled threshold which

deflects falling leaves and reduces the build-up of debris (*Hobbs & Brehme, 2017*). The HALT system has demonstrated almost perfect detection probability for small mammals, reptiles, amphibians and large invertebrates regardless of body size or temperature, and functioned satisfactorily in a field setting (*Hobbs & Brehme, 2017*). It was able to detect slow moving species but not those travelling at speed ($\geq 1$ ms$^{-1}$), and falling debris intercepting the optical beam was a cause of false triggers (*Hobbs & Brehme, 2017*). In another study where the utility of cameras to sample arthropods was compared to pitfall trapping, fallen debris was also found to lower arthropod detectability (*Collett & Fisher, 2017*). This may pose a more serious limitation on the efficacy of camera traps in heavily vegetated habitats, such as forests, heathlands or tall grasslands. Moreover, it presents a larger barrier for accurate artificial recognition software, in contrast to desert environments where there is greater contrast between invertebrates and the relatively uniform sand background.

Another downside of the HALT system or using a structure such as a drift fence is that they rely on fauna passing along a narrow trail or pathway, and therefore would not be as practical in open habitats such as areas of bare sand. Fauna may also be less inclined to pass up and over the artificial threshold of the HALT system, although the use of a lure or bait may overcome this issue. Finally, the HALT camera battery lasts only one to two weeks (*Hobbs & Brehme, 2017*). An improved battery operating system may be required, such as greater battery capacity or solar charging, so that a longer monitoring period could be achieved. Reduction in the frequency of images to every 15 min may reduce the number of images or extend the sampling timeframe without decreasing the effectiveness of cameras (*Collett & Fisher, 2017*).

## Future directions

Investigation into the efficacy of camera traps as a tool for uncovering the diel and spatial activity patterns of arthropods presents a significant case for the importance of technological advances in ecology and for gaining a deeper understanding of the biology of understudied organisms. Despite the limitations discussed above, a relatively long-term dataset was gathered, i.e., over a 3-month period rather than direct observations from a discrete time period, such as from a single field trip. If camera technology continues to improve (e.g., the HALT system, automated identification), the application of cameras as a cost-effective, non-invasive tool to study invertebrates, particularly those of conservation interest, would be invaluable. It would permit detailed research into general invertebrate behaviour and biology, as well as enable a greater understanding of the role of invertebrates in terrestrial ecosystems. It would also enable insight into trophic interactions and community-level processes, help inform us as to how invertebrates may be impacted by environmental fluctuations such as climate change and, more broadly, could be applied in the study of landscape health and function, as invertebrates are often used as indicators of habitat restoration and rehabilitation (*Fagan et al., 2010*; *Lenhard & Witter, 1977*; *Orabi, Moir & Majer, 2010*). There could also be economic benefits as awareness of invertebrate crop pest activity may result in more targeted treatments, for example, native predator abatement (*Kuusk & Ekbom, 2012*; *Kuusk et al., 2008*; *Lavandero et al., 2004*; *Von Berg, Traugott & Scheu, 2012*).

## CONCLUSIONS

Cameras with time-lapse settings have considerable utility as a tool for revealing the diel and spatial activity patterns of arthropods and for broader ecological investigations and long-term monitoring of invertebrates.

## ACKNOWLEDGEMENTS

We acknowledge the Wangkamadla people as the Traditional Owners of Ethabuka Reserve. We recognize and respect the enduring relationship they have with their lands and water, and we pay our respects to Elders past, present and emerging. We thank Bush Heritage Australia, in particular Alex Kutt and Matt and Amanda Warr, for allowing access to the study site, Bobby Tamayo for his valuable logistical assistance in the field, Glenda Wardle for helpful discussion, the many volunteers who assisted with data collection, and Larissa Potter and Graeme Finlayson for an earlier review of the draft manuscript.

### Funding

This work was funded by the Australian Research Council and the Australian Government's Terrestrial Ecosystem Research Network, an Australian research infrastructure facility established under the National Collaborative Research Infrastructure Strategy and Education Infrastructure Fund—Super Science Initiative through the Department of Industry, Innovation, Science, Research and Tertiary Education. This research also received support from the Australian Government's National Environmental Science Program through the Threatened Species Recovery Hub. Christopher R Dickman was also supported by an Australian Research Council Fellowship. The funders had no role in study design, data collection and analysis, decision to publish, or preparation of the manuscript.

### Grant Disclosures

The following grant information was disclosed by the authors:
Australian Research Council.
Australian Government's Terrestrial Ecosystem Research Network.
Australian research infrastructure facility established under the National Collaborative Research Infrastructure Strategy and Education Infrastructure Fund—Super Science Initiative through the Department of Industry, Innovation, Science, Research and Tertiary Education.

### Competing Interests

The authors declare there are no competing interests.

### Author Contributions

- Tamara I. Potter conceived and designed the experiments, performed the experiments, analyzed the data, prepared figures and/or tables, authored or reviewed drafts of the paper, and approved the final draft.

- Aaron C. Greenville conceived and designed the experiments, analyzed the data, authored or reviewed drafts of the paper, and approved the final draft.
- Christopher R. Dickman conceived and designed the experiments, performed the experiments, authored or reviewed drafts of the paper, and approved the final draft.

### Field Study Permissions

The following information was supplied relating to field study approvals (i.e., approving body and any reference numbers):

There is a Collaborative Research Agreement between the University of Sydney Desert Ecology Research Group and Bush Heritage Australia, who manage the reserve. Alex Kutt (BHA Regional Ecologist) and Matt and Amanda Warr (Ethabuka Reserve managers) provided permission at the time of the study.

### Data Availability

Updated invertebrate and camera data, spotlighting data, and R code are available in the Supplemental Files.

### Supplemental Information

Supplemental information for this article can be found online at http://dx.doi.org/10.7717/peerj.10684#supplemental-information.

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
