# Peer review of "Night of the hunter: using cameras to quantify nocturnal activity in desert spiders"

_PeerJ, doi:10.7717/peerj.10684_

## Round 0.1 · original submission · Major Revisions

After reading through this manuscript and receiving two reviews, it is clear that there are some good points to be made regarding the use of camera traps for invertebrate sampling, however several critical issues have been raised that require diligent attention if this manuscript is to be considered further for publication.

Reviewer 1 gave a very positive review and has made a number of thoughtful suggestions for changes, as well as additional information that could be added to enhance the manuscript’s scope. Reviewer 1 also brought up some important methodological issues regarding the application of day vs night surveys and what information can be gleaned from these comparisons. This is a good point and this issue should be addressed (see review section “validity of the findings” for details) This is not a major issue and one that can likely be address by adjusting the interpretations made, and then the text accordingly.

Reviewer 2, on the other hand, found some other - somewhat more troubling - issues with the manuscript that I agree need to be taken very seriously. These issues pertain to: (1) transparency with regards to reusing datasets, (2) careful attention to avoid self-plagiarism and duplicate publication, and (3) reanalyzing your datasets using assumptions based on taxon appropriate time-frames.
1) Transparency: Although it is perfectly acceptable to reused previously published datasets to test new questions, or even just applying new analyses, it should be stated clearly (e.g., “Previous research has seen x, y, and z [citation], and we reanalyzed this data to test a, b, and c.”) and with direct links to the original datasets. It is important to be open and transparent and ensure the language you are using does not imply that this data was solely collected for the purpose of testing the questions put forward here, but rather was a post-hoc analysis of data from a previous study. Honesty and transparency are key.

2) Self-plagiarism and duplicate publication: Between Lines 123-192 about half of the wording and sentences are almost identical to your previous publication (Potter et al. 2018; https://royalsocietypublishing.org/doi/10.1098/rsos.171872). This is of course not acceptable. Although I understand that the information being conveyed is going to be generally the same, the stating of this information MUST be unique. This issue is two-fold, in it that the authors should be avoiding self-plagiarism, but also so that this journal is not publishing duplicated content (especially since that content is already licensed to another publisher). This, in some opinions, would be grounds for pretty much immediate rejection – but I am assuming it was an oversight on the authors part, rather than an intentional act. Please, and with line by line attention, take careful and attentive care to ensure all content in the manuscript is original. This includes text (see annotated pdf with highlighted lines, this is not a complete assessment, merely what a Reviewer caught during their reading – the onus is on you to find any and all replicated text) and figures (Figure 2 is identical to Figure 1 in the Supplementary Material of Potter et al. 2018, and parts of Figure 3 (here) are identical to Figure 2 from Potter et al. 2018). This issue with the figures is particularly in poor form. It is obviously a “cut and paste” action that is completely unacceptable. There are many ways to present data, and I would strongly recommend that the authors find ANOTHER method of illustrating diel activity patterns (other than the radial time method used here and previously by the authors) to avoid any further confusion and the issues of self-plagiarism and duplicate publication. Overall, this issue is gravely serious, and I trust the authors will pay careful attention to this, as well and endeavor to avoid it in all future works.

3) Reanalysis: It is important that the assumption used in sorting and analyzing one’s data be taxa and context appropriate. One of the reviewers observed that the time intervals between your fixed time camera trap (every 5 minutes) was not a taxa appropriate time-frame to conclude that it was a unique individual. In the manuscript you state “To ensure independence, multiple photographs likely to be of the same individual (photographs in sequence <2 min apart) were removed prior to analysis (Greenville et al. 2014; Heavener et al. 2014).” The references you cite for this timeframe are both mammal-based, whereas other studies looking specifically at invertebrates (including spiders) saw that the mean time an individual stayed in frame was over 30 mins (Collett & Fisher 2017; https://onlinelibrary.wiley.com/doi/pdf/10.1002/ece3.3275). This is a large discrepancy (between 2 and 30 minutes), which no doubt would alter the dataset if properly adjusted. I would recommend that during the revision the authors alter there cut off for grouping photographs from the mammal-centric 2 min to the invert appropriate 32 min (using the Collett & Fisher (2017) mean time). So that any two photos captured within a 32 min window be grouped as the same individual. For example, this would mean that over the course of a night photos at 18:10, 18:40, 19:05, 19:15, 19:30, 20:00, and 20:20 would all be counted as the same individual, since each photo is less than 32 min from the previous. This can be altered with the caveat that clearly different spiders (e.g., having noticeable size differences or with unique markings or missing limbs) could be accounted for as unique within the 32 min period. This will no doubt be a fair amount of effort, however it will yield a much higher likelihood of generating accurate information that is taxon specific, and is a better option than using mammalian time-frame to infer invertebrate movement patterns.

Overall, the idea of this paper is not flawed and both reviewers have provided excellent comments on how it can be improved. However, the manner in which it is presented (issues # 1 & 2) and some of the assumptions made (issue # 3 and Reviewer 1’s comment on day vs night surveys) must be addressed if this manuscript is to be considered further. For obvious reasons issue # 2 is the most important to rectify, but issues # 1 and 3, and well as Reviewer 1’s comments are not trivial either and all should be addressed to the fullest extent. Please do so before resubmitting.

Reviewer 1 ·

Basic reporting

I really like this paper, it is clear, well laid out and will be of significant interest to the entomological community (where the use camera traps to observe invertebrates have long been dreamt of!)
The background is well researched and the study is well placed in the Simpson dessert as it presents an interesting proof of concept for the camera traps and utilises relevant environmental variation (eg. responses to fire, which is a "hot" topic in Australia at the moment).

Specific comments

Intro:
• L62: if you want a reference on the importance and use of invertebrate trait data for international comparisons Wong M.K.L., B. Guénard & O.T. Lewis 2019. Trait-based ecology of terrestrial arthropods. Biological Reviews 94:999-1022. doi:10.1111/brv.12488
• L64. Instead of destructive like this to be more clear. Eg. Usually result in the death of the individuals captured

Methods:
• Please include a section in the methods on how you identified lycosid spiders from other similar spiders in both the spotlighting and the images

Figures:
• Not sure why there are two figure captions for the figures? I guess this is a result of the manuscript input forms
• Figure 1: can you add burned unburned labels to the figure?
• Figure 1 legend, the current wording is a little confusing. consider rephrasing to: Two camera positions were tested: a) angled at 45 degrees (orange squares n=6) etc.
• Figure 3: is there any way to make the frequency observation axis more clear? You need to add % to the axis too, as in Figure 4. It’s a bit unclear why axis has the points 3.75 and 7.5 rather than regular the points on other figures (same for a in Figure 4). If they were all the same eg. 5%, 10%, 15%, 20% it would be easier to make comparisons. Figure 5 looks great on all these points, just make Figure 3 and 4 conform to the same details.

Discussion:

I'd like to see mention of the other data that can be collected from camera traps. Specifically useful trait data. One of the interesting life history traits of wolf spiders is that they carry their offspring on their backs. Was there any way of detecting this in the photos? This would provide additional (and very useful) data (eg could study differences in reproductive cycles in different areas or boom bust cycles). I’d also like a discussion of what types of trait data eg. Size, maybe even speed or activity could be collected from the photos.

The discussion seems to focus mainly on the differences in activity patterns, but the number of spiders detected (eg. The number of photos) deserves some discussion too (eg paragraph 3 on dune vs crest activity, where there was a difference in number of individuals detected, but no difference in activity patterns).

Experimental design

The experimental set up is good, and I like that it addresses two different types of environmental variation (the location on the dune and the time since fire).

I also think that the comparison between the spotlight data and the camera trap data is useful, despite the limitations (which are well covered in the discussion).

Validity of the findings

I might have misunderstood here, but if the lycosids were only found in the time laps, and you didn’t use the time laps during the day then I don’t think you can make the claim that Lysocids are only nocturnal. Same problem with the spotlighting, unless you also did transects during the day you cant make the claim they weren’t active then. In other habitats lycosids are known to be quite active during the day.

Desert habitats are great in that they have nice open spaces in which to face a camera and get a good contrast of the spider against the sand. You mentioned this briefly in the discussion, but could you comment more on how you think this method would work in different ecosystems? Eg. Against a forest floor or in grasses? I think this might be a large limitation.

Additional comments

I like your coverage of the limitations, but of course, there are many. My thoughts below:

What do you think the potential is for biodiversity studies? You may not have this data, but it would be interesting to know the other invertebrate fauna that were also detected (this could widen the scope of the paper to appeal to other invertebrate researchers). Eg. Did any invertebrates trigger the motion capture? This has been the main problem in using this method in the past.

What do you think of the possibility of using baiting the stations (eg tethered prey like meal worms) to attract arthropod predators?

Reviewer 2 ·

Basic reporting

The dataset reported here does not appear to be "self-contained."

I took a look at Potter et al. 2018 (https://royalsocietypublishing.org/doi/10.1098/rsos.171872) because I was interested in seeing how the two Lycosa species had been identified and this paper was referenced in line 144. Upon closer examination I found that the dataset reported in the manuscript currently under review in PeerJ has already been published in Royal Society Open Science as part of a paper investigating space use and activity by wolf spiders and dunnarts.

The current paper clearly has a different goal and adds some additional analyses to the findings already reported in Potter et al. 2018, but nowhere in the manuscript is it mentioned that these data have already been published elsewhere. Rather, the current paper is written as if the data were collected specifically to ask the questions (1-3) stated in the abstract, rather than reporting that these data were collected as part of a larger study that used camera traps to detect dunnarts and spiders, and this is seriously misleading.

I have attached copies of the pdfs of the manuscript under review and Potter et al. 2018, in which identical or near-identical text from the methods and results section are highlighted. Figure 2 is identical to Figure 1A in the supplementary material of Potter et al. 2018, and Figure 3b. is indentical to figure 2Ab (https://royalsocietypublishing.org/doi/suppl/10.1098/rsos.171872). This is also an issue of self-plagiarism.

Experimental design

Please be explicit that the cameras were deployed and data collected as part of a broader study.

Validity of the findings

In lines 165-166 the authors state that “To ensure independence, multiple photographs likely to be of the same individual (photographs in sequence <2 min apart) were removed prior to analysis (Greenville et al. 2014; Heavener et al. 2014).”

This method of determining independence is seriously flawed.

First, this threshold appears to be based on one used by Greenville et al. for dingoes, and the other cited paper was on rats. These are mammals that trigger camera traps and no doubt two minutes makes sense for these animals.

A study which used camera traps to study spiders and other arthropods (Collett & Fisher 2017; https://onlinelibrary.wiley.com/doi/pdf/10.1002/ece3.3275), on the other hand, found that individual arthropods captured in successive frames of time lapse images stayed in the frame for a mean of 32 +/- 13 min, and that one spider remained in view of a camera for over 15 h.

The current paper found that spiders never triggered motion sensing on the camera traps, but rather spiders were only ever captured in time lapse photographs. These photos were taken every 5 minutes (lines 135-136) so by definition no photographs of spiders can be excluded because they are <2 min apart.

When one examines the raw data submitted with this paper, it becomes clear that a large number of presumed independent photographs of spiders are successive frames taken at 5 min intervals. For instance, in the first 36 lines of data, there are 4 successive frames from 22:20-22:35 on 2016-07-18, 27 successive frames from 4:35-6:55 on 2016-09-01, and 5 successive frames from 6:35-6:55 on 2016-09-03.

It seems much more likely that these 36 images (used as independent data points in analysis) represent three unique individuals rather than 36. Since the raw image data are not included with this submission I cannot check whether successive images appear to contain the same species or same individual, or how much the spiders move from frame to frame. However, given what I know about the ecology and behaviour of wolf spiders (they frequently remain still while hunting in a relatively small area, and also see Collett & Fisher 2017, mentioned above) it seems much more likely that the same individuals were frequently photographed multiple times rather than new individuals happening to enter the frame approximately every five minutes. Indeed, it seems incredible to me that a single camera (camera 1) deployed on 7 July detected no spiders until 18 July, then detected 4 spiders in rapid succession, then no spiders until 1 September, and 27 spiders in a row on that night, and then 5 spiders two nights later.

If these likely repeated records of the same individuals were taken into account in analyses this would not be a fatal flaw, but as the manuscript currently stands this is a major issue.

Annotated reviews are not available for download in order to protect the identity of reviewers who chose to remain anonymous.

---

## Round 0.2 · Minor Revisions

I would like to thank the authors for taking the reviewers previous comment to heart and putting in a solid effort in revising their manuscript. It is reading much better now. Well done!

The only reason that it has not been accepted outright is a small matter of including the two points raised by Reviewer 2 (see below). The first is including information on how you ID'd the wolf spiders. And the second is trying to standardise your figure style between Figures 4, 6, and 7. These figures are using a very similar style to present data, but not quite the same - which makes it look a bit off. These seem like pretty small items and I am sure you will be able to address them and have it back to us very quickly. After this, I have no doubt this paper will be accepted.

I would remind the authors to (in this last round of edits) have a very careful read-through of the manuscript, just in case any typos have slipped their (and our) gaze.

Reviewer 1 ·

Basic reporting

I've had a good read through the rebuttal and the new manuscript. The authors did a very good job of addressing all of my comments and the paper reads really well now.

Experimental design

I'm happy with the responses the authors have made to previous methodological concerns.

Validity of the findings

no comment

Additional comments

This is a valuable contribution to the literature and reads very well, nice work

Reviewer 2 ·

Basic reporting

I have only one comment:

Please describe how you identified the spiders to the genus Lycosa (cite key used, for example—I could not find any such reference in the previous paper about this dataset) and whether voucher specimens have been deposited in a museum collection (if they have not been, please do so, so that future researchers can verify the identification and benefit from the knowledge of which taxa are present at the study site).

Experimental design

Good.

Validity of the findings

Good.

Additional comments

This is an interesting paper containing valuable data and I enjoyed reading the updated version. I thank the authors for their efforts to deal with my comments and those of the other reviewers.

Besides the addition of reference(s) for the spider identification and information about voucher specimens above, I have only one other comment:

Why are the circular data displayed differently from figure 4 in figures 6 and 7? Either way seems fine but the latter two don’t seem to indicate the mean and consistent data display would be preferable.

I think that the manuscript is very clear and much improved based on the revisions, and is a valuable contribution.

---

## Round 0.3 · accepted · Accept

Thank you for tackling those last two reviewer comments - and doing so so promptly. This is a great paper and a neat study and I am very pleased we are able to accept it for publication. Well done and cheers!